# Efficacy and safety of acupuncture for postpartum hypogalactia: A systematic review and meta-analysis of randomized controlled trials

Qiong-Nan Bao [1,2,3]☯, Zi-Han Yin[3]☯, Yuan-Fang Zhou[3], Ya-Qin Li[3], Xin-Yue Zhang[3], Man-Ze Xia[3], Zheng-Hong Chen[3], Wan-Qi Zhong[3], Jin Yao[3], Ke-Xin Wu[3], Zhen-Yong Zhang[1,2], Shao-Jun Xu[1,2]*, Fan-Rong Liang [3]*

1 Department of Traditional Chinese Medicine, The First People's Hospital of Yunnan Province, Kunming, Yunnan, China, 2 The Affiliated Hospital of Kunming University of Science and Technology, Kunming, Yunnan, China, 3 School of Acu-Mox and Tuina, Chengdu University of Traditional Chinese Medicine, Chengdu, Sichuan, China

☯ These authors contributed equally to this work.
* acuresearch@126.com (FRL); xinyun1220@126.com (SJX)

**Data Availability Statement:** All relevant data are within the manuscript and its Supporting information files.

## Abstract

### Background

Postpartum hypogalactia (PH) is prominent during lactation and may negatively impact the mother's or infant's health. Acupuncture is widely used to increase maternal breast milk production. However, the effects of acupuncture on PH remain unclear. Therefore, this review aimed to evaluate the efficacy and safety of acupuncture in individuals with PH.

### Materials and methods

Articles on potentially eligible randomized controlled trials (RCTs) on acupuncture for PH published from database inception to October 2023 were retrieved from the PubMed, Web of Science, Cochrane Library, EMBASE, EBSCO, Scopus, China National Knowledge Infrastructure, Chinese Biomedical Literature Database, WanFang, and VIP databases. Two reviewers independently screened the records, extracted essential information, and evaluated the methodological quality of the RCTs using the revised Cochrane risk-of-bias (RoB) tool. The primary outcome was a change in serum prolactin (PRL) levels before and after treatment. Secondary outcomes included milk secretion volume (MSV), total effective rate (TER), mammary fullness degree (MFD), and exclusive breastfeeding rate (EBR). Meta-analyses were performed using RevMan v5.4. Finally, the quality of evidence was evaluated using the Grading of Recommendations, Assessment, Development, and Evaluation tool.

### Results

This study included 19 RCTs involving 2,400 participants. The included studies were classified as having an unclear to high RoB. Our findings indicated that, overall, acupuncture showed a significant effect in increasing serum PRL levels (standardized mean differences

**Funding:** This study was financially supported by the "Central Financial Transfer Payment to Local Projects in 2022 of National Administration of Traditional Chinese Medicine", which does not have a specific grant number. Fan-Rong Liang was the study funder.

**Competing interests:** The authors have declared that no competing interests exist.

[SMDs] = 1.09, 95% confidence interval [CI]: 0.50, 1.68), MSV (SMD = 1.69, 95% CI: 0.53, 2.86), TER (relative risk [RR] = 1.25, 95% CI: 1.10, 1.42), and EBR (RR = 2.01, 95% CI: 1.07, 3.78) compared to that in the control group; however, no difference in MFD (SMD = 1.17, 95% CI: −0.09, 2.42) was observed. In the subgroup analysis, acupuncture combined with Chinese herbs or conventional treatment was significantly more effective in increasing serum PRL levels, MSV, and TER than did Chinese herbs or conventional treatment alone. Moreover, acupuncture alone resulted in significantly higher serum PRL levels compared to Chinese herbs; however, this benefit was not observed for TER and MFD. The quality of evidence was critically low.

## Conclusion

Acupuncture may effectively increase milk secretion in women with PH. However, owing to the low quality of evidence, further rigorously designed studies are warranted to confirm our findings.

## Introduction

Postpartum hypogalactia (PH) is a common condition characterized by complications in producing sufficient milk during lactation [1]. The World Health Organization recommends exclusive breastfeeding for the first 6 months after childbirth [2]; however, only approximately 40% of infants younger than 6 months are exclusively breastfed [3]. The global exclusive breastfeeding rate is low at 33% [4], with approximately 29.2% and 25% in China and Europe, respectively [5, 6]. PH is the most frequently reported reason for discontinuing breastfeeding [7–9]. Human milk, ideally suited for infant nutrition, protects children against infectious diseases in infancy and chronic disorders later in life [10–12]. Breastfeeding can reduce the risks of breast and ovarian carcinomas, diabetes, hypertension, and heart disease in mothers [13–15]. Moreover, exclusive breastfeeding offers significant economic benefits for families and the society. A 10% increase in exclusive breastfeeding for up to 6 months or continued breastfeeding for up to 1 or 2 years could potentially save at least $312 million in medical costs in America and $7.8 million in the United Kingdom in 2012 [16]. Despite these advantages, several mothers cannot breastfeed their children due to PH. Thus, helping women with PH augment their milk supply can increase breastfeeding rates, thereby improving maternal and child health and reducing healthcare costs.

PH is associated with physiological, psychological, and emotional factors [17] and can be treated using various strategies. A 2020 Cochrane review by Foong et al. [18] concluded that oral galactagogues may increase milk volume and infant weight; however, adequate supporting evidence is lacking. Commonly used drugs, such as domperidone, metoclopramide, and sulpiride, have limited short-term effects on breast milk production and are associated with multiple adverse reactions [19–22]. Other natural galactagogues, such as fenugreek and silymarin, lack definitive or consistent scientific evidence to support their contribution to increased milk production [23]. Similarly, other management options, including skin-to-skin therapy, breast pumps [24], breastfeeding counselling [25], early education, and timely support [26], seem unsatisfactory. Therefore, identifying additional, safe, and effective therapies for treating PH is necessary.

Acupuncture is an ancient form of traditional Chinese medicine (TCM), and its popularity has increased worldwide over the last several decades [27]. As a complementary and alternative therapy, acupuncture is a frequently recommended treatment for PH in European countries [28]. According to the TCM theory, acupuncture treatment involves inserting needles into acupoints along the meridians to enhance Qi and blood flow, achieving yin and yang balance throughout the body and promoting overall health [29]. Furthermore, emerging evidence suggests the potential therapeutic benefits of acupuncture on PH. Liu et al. [30] observed higher serum prolactin (PRL) levels, increased breast fullness, and greater lactation in the acupuncture group compared with that in the controls. Maged et al. [31] reported that electroacupuncture significantly increased postnatal milk secretion and infant weight in primiparas. Although several randomized controlled trials (RCTs) [30–32] have reported that acupuncture can effectively treat PH, no systematic review has been published to support this notion. Thus, this study aimed to determine the clinical efficacy and safety of acupuncture in women with PH.

## Materials and methods

This systematic review was registered in the PROSPERO registry (CRD42022351849), and the final report has been presented following the Preferred Reporting Items for Systematic Reviews and Meta-Analyses (PRISMA) statement (S1 File) [33]. The detailed protocol has been peer-reviewed and published [34].

### Search strategy

Electronic databases, including PubMed, Web of Science, Cochrane Library, EMBASE, EBSCO, Scopus, China National Knowledge Infrastructure, Chinese Biomedical Literature Database, WanFang, and VIP, were searched for RCTs published from database inception to October 2023. The search terms used were ("acupuncture" OR "electroacupuncture") AND ("lactation" OR "hypogalactia" OR "breastfeeding") AND ("randomized controlled trial"). Chinese databases were searched using the Chinese translations of these terms. The detailed search strategies for each database are presented in the S2 File. Additionally, the references of published reviews were manually examined, and grey literature was identified by searching the clinical trial registry platform. An updated search using the same strategy was performed on October 19, 2023. Two reviewers (YF-Z and SJ-X) independently screened the titles and abstracts of all search results for potentially eligible studies and determined the final selection after reading the full text. Discrepancies were resolved through group discussion with a third reviewer (ZH-Y).

### Eligibility criteria

**Types of studies.**  RCTs published in English or Chinese were included without limitations on publication type. However, non-RCTs, repeated publications, case series, reviews, literature without the full text available, and experimental animal studies were excluded.

**Types of participants.**  Women with PH were included, regardless of age, disease duration, or parity. However, participants with other postpartum breast disorders, such as breast engorgement, mastitis, and breast abscess, were excluded.

**Types of interventions.**  Participants in the treatment group received manual acupuncture or electroacupuncture without limitation on the acupoints, needle materials, treatment frequency, or duration. Interventions not involving needle insertion, such as acupressure or moxibustion, were excluded.

**Types of control groups.**  In the control group, participants receiving sham acupuncture; Chinese herbs (CHs); or conventional treatment (CT), including breast sucking, postpartum

routine care, and breastfeeding education; were included. Studies that used interventions, such as drugs, galactagogues, massage, or any other complementary therapy, in the control group were excluded.

**Types of outcomes.** The primary outcome was a change in serum PRL levels from baseline to post-treatment. Secondary outcomes included milk secretion volume (MSV), total effectiveness rate (TER), mammary fullness degree (MFD), exclusive breastfeeding rate (EBR), and adverse events.

## Data extraction

Two reviewers (YF-Z and YQ-L) independently extracted and recorded the following data from the included studies: participant characteristics (age, parity, delivery mode, and disease duration) and study characteristics (author name, publication year, sample size, intervention mode, control mode, and outcome index). If published data were inadequate, the corresponding authors were contacted for additional information.

## Risk-of-bias assessment

Two reviewers (YQ-L and XY-Z) assessed the methodological quality of the included trials using the revised Cochrane risk-of-bias (RoB 2.0) tool [35], which assesses five distinct domains: 1) randomization process, 2) deviations from the intended interventions, 3) missing outcome data, 4) outcome measurement, and 5) selective reporting. The bias risk of each domain was classified as high risk, some concern, and low risk, contributing to the overall RoB assessment. Conflicting opinions were resolved through discussion with a third reviewer (ZH-Y).

## Statistical analysis

A meta-analysis was performed using the RevMan software V.5.4 (Cochrane Collaboration, London, UK). Dichotomous data were calculated using relative risk (RR) with 95% confidence intervals (CIs). Continuous data were analyzed using standardized mean differences (SMDs) with 95% CIs. A heterogeneity test was performed using $I^2$ statistics. $P < 0.05$ was considered statistically significant per the Cochrane Handbook [36]; $I^2 > 50\%$ indicated significant heterogeneity, and a random-effects model was adopted. Subgroup analyses were performed based on different comparators and treatment durations to explore potential sources of heterogeneity. Sensitivity analyses were performed by individually removing each study to verify the stability of the results and further interpret the heterogeneity. Moreover, publication bias was evaluated using funnel plots and Egger's test.

## Quality of evidence assessment

Evidence quality was independently evaluated by two reviewers (QN-B and YF-Z) following the Grading of Recommendations Assessment, Development, and Evaluation (GRADE) tool [37], which evaluates the following eight aspects: limitations in study design, inconsistency, indirectness, imprecision, publication bias, large effect, dose-response, and all plausible confounding factors. The quality of the evidence was graded as high, moderate, low, or critically low. Disagreements were resolved through a discussion with a third reviewer (ZH-Y).

## Results

### Study selection

Out of the 3,199 articles identified from 10 databases and 2 from other sources, 1,383 duplicates were removed. Additionally, 1,735 studies were preliminarily eliminated through the initial screening of titles and abstracts. Finally, after reviewing the full texts of 81 selected papers, 62 were excluded (17 non-RCTs, 29 with no comparisons of interest, 4 ineligible intervention groups, 3 duplicates, 5 trial registration articles, 2 with no participants of interest, and 2 without full texts available), and 19 RCTs were retained. The PRISMA flowchart of the literature search is presented in Fig 1, and the excluded full-text studies with reasons for exclusion are detailed in S1 Table.

### Study characteristics

The included studies were from Italy [38], Thailand [32], and China and were published between 2000 and 2022. A total of 2,400 participants were examined, with 1,241 in the treatment groups and 1,159 in the control groups, and sample sizes ranged from 60–350. Participants with PH were aged 20–35 years. The course of the disease ranged from 2 days to 7 months. Eleven studies [32, 38–47] described the inclusion and exclusion criteria of participants. Twelve trials [39–42, 44–51] recruited women who delivered naturally or via cesarean section, and one study [46] enrolled women who delivered via cesarean section only. Six

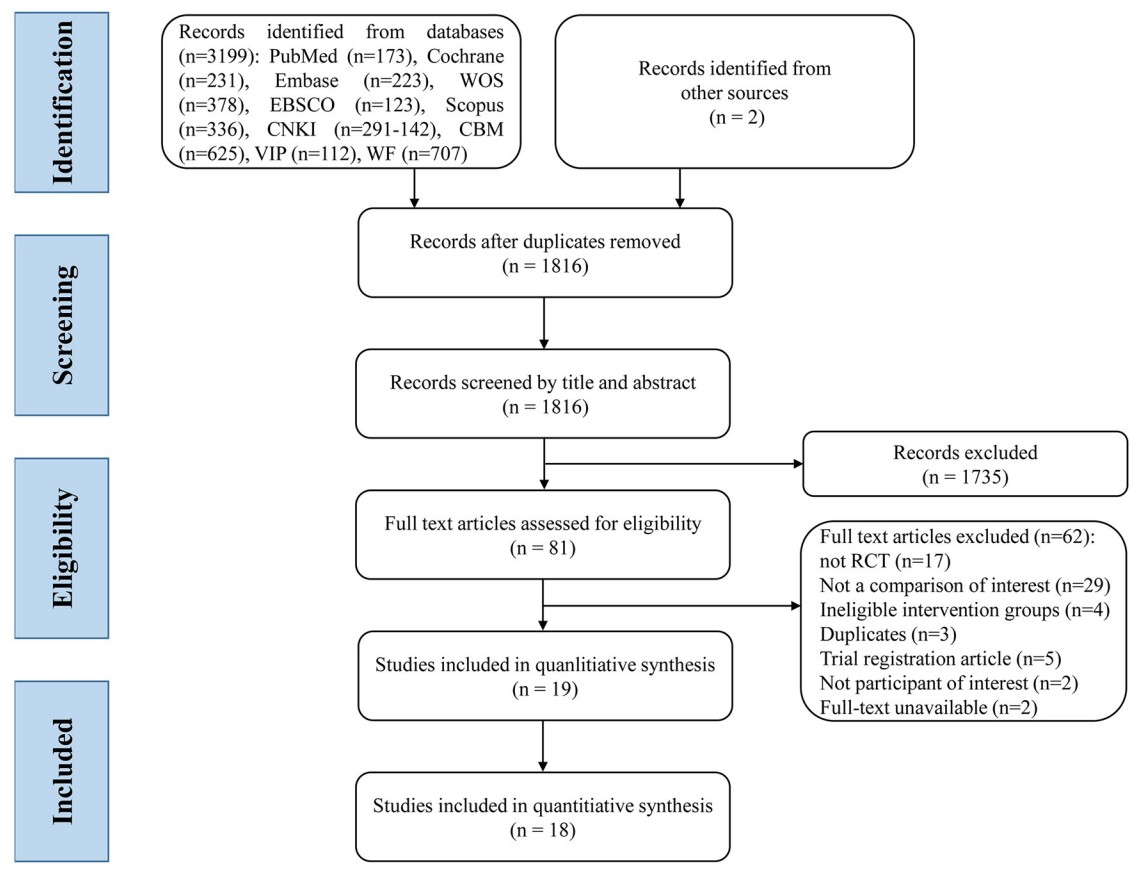

**Fig 1. Flowchart of the literature selection process and screening results.**

studies [40, 41, 44, 47, 52, 53] recruited primiparas and multiparas, and one study [39] enrolled only primiparas. Three studies [32, 40, 47] mentioned the inclusion of women who delivered after 37 weeks of gestation. Regarding exclusion criteria, eight studies [39, 40, 42–47] excluded severe primary disease, seven [39, 40, 42–46] excluded psychotic disease, two [38, 39] excluded pregnancy complications, and five [32, 40, 42, 44, 47] excluded breast anomalies or other breast diseases. In 13 studies [32, 38, 40–44, 47, 49, 51–54], participants were classified into five TCM syndrome types based on TCM syndrome differentiation, which is a core principle of TCM in recognizing and treating diseases [55]. Seventeen studies were conducted using a two-group parallel design; seven [38, 44–47, 51, 54] compared acupuncture with CH or CT; ten [32, 39–43, 48, 49, 53, 56] compared the combination of acupuncture and CH/CT with CH/CT alone; two [46, 50] were conducted using a three-arm study design, with one study comparing acupuncture plus CH with acupuncture alone and CH alone, and the other comparing acupuncture with sham acupuncture and CT. The treatment duration in the included studies ranged from 3 to 14 days. Moreover, TER was the most frequently mentioned outcome. No study had pre-registered their trial protocol on a clinical registry platform. The main characteristics of the included RCTs are summarized in Table 1.

## Acupuncture details

Following the revised Standards for Reporting Interventions in Clinical Trials of Acupuncture (STRICTA) tool [57], details of the acupuncture methods were extracted and are presented in Table 2. The number of needle insertions per participant per session in the 19 trials ranged from 1 to 25. The most frequently used acupoints for PH were Danzhong (CV17), Shaoze (SI1), Rugen (ST18), and Zusanli (ST36), whereas the most commonly used needle stimulation method was manual acupuncture. Nine studies [32, 38, 44, 45, 49–51, 54, 56] reported that participants experienced the Deqi sensation. The most frequently used acupuncture brand was Hwato, and the most common diameters and lengths of the needling instruments were 0.2 and 1.5 mm, 0.25 and 40 mm, and 0.30 and 25 mm. The number of treatment sessions varied; the most common frequency was daily sessions for 7 days. For the other components, more than half of the included trials covered the details of other interventions. Regarding the practitioners, only two studies [32, 38] included the details of the acupuncturists.

## Risk of bias

Three studies were classified as having a "high RoB" in domain 1 (randomization process): two of them [51, 54] reported an inappropriate method for sequence generation, and one [52] suggested a baseline imbalance between the intervention groups due to an unbalanced participant allocation ratio (62:34). Four of the included RCTs [38, 42, 45, 46] were classified as having a "low RoB" in domain 1 because they were conducted with a concealed allocation. All studies were evaluated as having "some concern" for RoB in domain 2 (deviations from the intended interventions) because the participants and personnel were not blinded. All studies were assessed as having a "low RoB" in domain 3 (missing outcome data). Three trials [32, 38, 44] were rated as having a "low RoB" in domain 4 (outcome measurement), whereas the remaining were seen as having "some concern" for RoB. In domain 5 (selective reporting), 14 studies were rated as having "some concern" for RoB, of which 8 [32, 38, 40, 43, 47, 49, 51, 53] reported results inconsistent with the outcome measurements, and 11 did not report the selection criteria [48–54, 56] or treatment duration [39, 43, 48]; the rest were rated as having a "low RoB." Overall, most studies were assessed as having "some concern" for RoB, except for three studies rated as having a "high RoB." The RoB is displayed in Figs 2 and 3.

**Table 1. Characteristics of the included studies.**

| Study | Country | Language | Sample Size (A/B/C) | Diagnostic criteria | Inclusion criteria | exclusion criteria | TCM syndrome differentiation | Age (year) | Parity (case) | Delivery mode | Disease duration (day/month) | (A) Experimental group | (B) Control group I | (C) Control group II | Treatment duration | Efficacy and safety criteria | Main results |
|---|---|---|---|---|---|---|---|---|---|---|---|---|---|---|---|---|---|
| Li 2022 | China | Chinese | 80 (40/40) | NR | Y | Y | NR | A: 28.66 ± 3.05 B: 28.58 ± 3.12 | Pp | A: NL 24, Cr 16 B: NL 22, Cr 18 | NR | MA + CT | CT | NR | NR | 1. PRL 2. EBR | 1. A>B 2. A>B |
| Sawittri Suwikrom, 2021 | Thailand | English | 60 (30/30) | NR | Y | Y | I | 29.72 ± 5.96 | NR | NR | NR | MA + CT | CT | NR | 3 d | 1. MSV 2. AE | 1. A>B 2. A = B |
| Qiu 2020 | China | Chinese | 70 (35/35) | ① | Y | Y | I | A: 31.83 ± 4.37 B: 30.47 ± 4.06 | A: Pp 21, Mp 14 B: Pp 19, Mp 15 | A: NL 23, Cr 12 B: NL 20, Cr 14 | A: 5.31 ± 2.17 d B: 4.94 ± 2.23 d | MA + CT | CT | NR | 7 d | 1. MSV 2. MFD 3. TER | 1. A>B 2. A>B 3. A>B |
| Yang 2020 | China | Chinese | 82 (41/41) | NR | Y | Y | V | A: 28.25 ± 4.29 B: 28.14 ± 4.35 | NR | A: NL 20, Cr 21 B: NL 19, Cr 22 | A: 2.14 ± 0.50 d B: 2.36 ± 0.52 d | MA+ CH | CH | NR | 14 d | 1. PRL 2. MSV 3. TER | 1. A>B 2. A>B 3. A>B |
| Li 2019 | China | Chinese | 100 (50/50) | ⑦ | Y | Y | I | A: 29.02 ± 5.17 B: 28.58 ± 4.45 | NR | A: NL 31, Cr 19 B: NL 28, Cr 22 | NR | MA + CH | CH | NR | 14 d | 1. PRL 2. MSV 3. MFD 4. TER | 1. A>B 2. A>B 3. A>B 4. A>B |
| Zhao 2018 | China | Chinese | 100 (50/50) | ①④ | Y | Y | III | A: 26.43 ± 3.11 B: 25.54 ± 2.59 | A: Pp 40, Mp 7 B: Pp 38, Mp 10 | A: NL 30, Cr 17 B: NL 34, Cr 14 | A: 6.40 ± 4.58 d B: 6.33 ± 4.25 d | EA | CH | NR | 7 d | 1. PRL 2. TEF 3. MFD 4. MSV | 1. A>B 2. A>B 3. A<B |
| Chen 2017 | China | Chinese | 140 (70/70) | NR | Y | Y | III | A: 30.32 ± 3.21 B: 31.32 ± 3.21 | NR | NR | A: 13 d B: 14 d | MA + CH | CH | NR | NR | 1. TER | 1. A>B |
| Liu 2017 | China | Chinese | 120 (60/60) | NR | NR | NR[52,53] | NR | 28.03 ± 5.47 | NR | NL 90, Cr 30 | NR | MA + CH | CH | NR | NR | 1. TER 2. AE | 1. A≥B 2. A<B |
| Xian 2017 | China | Chinese | 116 (58/58) | ① | NR | NR | NR | A: 26.5 ± 4.6 B: 27.8 ± 3.2 | NR | NR | NR | MA + CT | CT | NR | 14 d | 1. PRL 2. MSV 3. MFD | 1. A>B 2. A>B 3. A<B |
| Zheng 2015 | China | Chinese | 72 (35/37) | ① | Y | Y | III | A: 32.74 ± 4.49 B: 33.19 ± 4.24 | A: Pp 19, Mp 16 B: Pp 21, Mp 16 | A: NL 18, Cr 17 B: NL 22, Cr 15 | NR | MA | CH | NR | 7 d | 1. PRL 2. MSV 3. TER 4. AE | 1. A = B 2. A = B 3. A = B 4. A = B |
| Gao 2012 | China | Chinese | 60 (30/30) | ③ | NR | NR | I | A: 29.63 ± 2.08 B: 30.03 ± 2.09 | NR | A: NL 9, Cr 21 B: NL 11, Cr 19 | A: 15.87 ± 4.24 d B: 15.23 ± 3.75 d | MA + CH | CH | NR | 7 d | 1. MFD 2. TER | 1. A>B 2. A>B |
| Isabella Neri, 2011 | Italy | English | 90 (45/45) | NR | Y | Y | III | A: 34.9 ± 4.3 B: 33.9 ± 4.5 | NR | NR | NR | MA | CT | NR | 21 d | 1.EBR 2. AE | 1. A>B 2. A>B |
| Li 2010 | China | Chinese | 115 (58/57) | ① | NR | NR | II | A: 21–33 B: 21–34 | NR | NR | A: 5–11 d B: 5–12 d | MA | CT | NR | 10 d | 1. TER | 1. A>B |
| Zhang 2009 | China | Chinese | 240 (120/60/60) | NR | NR | NR | NR | A: 23–35 B: 22–36 C: 24–35 | NR | A: NL 100, Cr 20 B: NL 49, Cr 11 C: NL 50, Cr 10 | A: 5–60 d B: 7–60 d C: 6–60 d | MA + CH | MA | CH | 7 d | 1. TER | 1. A>B>C |
| Zhao 2008 | China | Chinese | 350 (224/126) | ① | NR | NR | III | A: 23.4 B: 24.2 | NR | A: NL 88, Cr 136 B: NL 70, Cr 56 | A: 4 d–7 m B: 4 d–7 m | MA | CH | NR | 5 d | 1. PRL 2. TER | 1. A>B 2. A = B |
| He 2008 | China | Chinese | 276 (138/138) | ① | Y | Y | NR | A: 28.99 ± 3.58 B: 29.53 ± 3.76 | NR | A: NL 53, Cr 85 B: NL 85, Cr 88 | NR | EA | CH | NR | 3 d | 1.PRL 2. MFD | 1. A = B 2. A = B |

(*Continued*)

**Table 1.** (Continued)

| Study | Country | Language | Sample Size (A/B/C) | Diagnostic criteria | Inclusion criteria | exclusion criteria | TCM syndrome differentiation | Age (year) | Parity (case) | Delivery mode | Disease duration (day/month) | (A) Experimental group | (B) Control group I | (C) Control group II | Treatment duration | Efficacy and safety criteria | Main results |
|---|---|---|---|---|---|---|---|---|---|---|---|---|---|---|---|---|---|
| Liu 2006 | China | Chinese | 96 (62/34) | ① | NR | NR | I, II, IV | A: 21–38 B: 20–36 | A: Pp 46, Mp 16 B: Pp 24, Mp 10 | NR | A: 10–50 d B: 8–48 d | MA | CH | NR | 10 d | 1. TEF | 1. A>B |
| Chen 2000 | China | Chinese | 98 (50/48) | ⑤⑥ | NR | NR | I | A: 28.7 B: 26.8 | Pp 92, Mp 6 | NR | A: 9 d–2 m B: 10 d–2.5 m | MA + CH | CH | NR | 14 d | 1. TER | 1. A>B |
| Zhang 2022 | China | Chinese | 135 (45/45/45) | NR | Y | Y | NR | A: 30.7 ± 3.9 B: 30.7 ± 4.1 C: 30.9 ± 3.8 | NR | Cr | NR | MA | SA | CT | 48 h | 1. MSV | 1. A>B, B>C |

*Note.* TCM: traditional Chinese medicine, NR: not recorded, Y: yes, ① Standards for Diagnosis and Curative Effect of Chinese Medical Symptom, ② Diagnostic Criteria for Gynecological Diseases, ③ Guidelines for the Clinical Research of Chinese Medicine New Drugs ④ Gynecology of Chinese Medicine, ⑤ Practical Chinese Gynecology, ⑥ The Complete Book of Chinese Acupuncture, ⑦ Diagnostic Efficacy Criteria of TCM Gynecological Disease, ⑧ Guidelines for the Diagnosis and Treatment of Common Gynecological Diseases in TCM; I: deficiency of Qi and blood syndrome, II: Liver-Qi stagnation syndrome, III: deficiency of spleen-Qi syndrome, IV: phlegm dampness and blood stasis syndrome, V: Qi deficiency and blood stasis syndrome, Pp: primipara, Mp: multipara, MA: manual acupuncture, EA: electroacupuncture, SA: sham acupuncture, CH: Chinese herb, CT: conventional treatment, NL: natural labor, Cr: Cesarean, d: day, m: month, h: hour, PRL: prolactin, MSV: milk secretion volume, TER: total effective rate, MFD: mammary fullness degree, AE: adverse events.

**Table 2. Details of acupuncture methods according to STRICTA.**

| Study | Acupuncture rational | | | Details of needling | | | | | | | Treatment regimen | | Other components | | Practitioner | Comparator interventions | |
|---|---|---|---|---|---|---|---|---|---|---|---|---|---|---|---|---|---|
| | 1a | 1b | 1c | 2a | 2b | 2c | 2d | 2e | 2f | 2g | 3a | 3b | 4a | 4b | 5 | 6a | 6b |
| Li 2022 | TCM | Y | Y | 5 | ST18, CV17, SI1 | NR | NR | Manual | 30 min | NR | NR | Twice a day | Y | NR | NR | NR | Y |
| Sawittri Suwikrom, 2021 | TCM | Y | Y | 25 | CV17, SI1, LR3, KI7, MI88.13, MI88.12, MI88.14, ML77.17, ML77.19, ML77.21, ML77.08, ML77.09, ML77.11 | 0.1-38mm | Deqi | Manual | 30 min | Diameter and length: 0.18mm & 25mm Needle brand: Maanshan Bond | 3 | Once a day, 3 days | Y | Y | Y | NR | Y |
| Qiu 2020 | TCM | Y | Y | 14 | RN12, RN10, RN6, RN4, ST24, ST25, SP15, KI13, Qipang (0.5 cuns far from RN6) | NR | NR | Manual | 30 min | Diameter and length: 0.25mm & 40mm Needle brand: NR | 7 | Once a day, 7 days | Y | NR | NR | NR | Y |
| Yang 2020 | TCM | Y | Y | 7 | CV17, SI1, ST36, LR3 | 7.5–37.5mm | NR | Manual | 20 min | Diameter and length: 0.30mm & 40mm Needle brand: Hwato | 14 | Once a day, 14 days | Y | NR | NR | NR | Y |
| Li 2019 | TCM | Y | Y | 4 | RN12, RN10, RN6, RN4 | NR | NR | Manual | 30 min | Diameter and length: 0.25mm & 40mm Needle brand: Huanqiu | 14 | Once a day, 14 days | Y | NR | NR | NR | Y |
| Zhao 2018 | TCM | Y | Y | 11 | ST18, CV17, SI1, ST36, GB21, 1point (0.5 cuns far from CV17) | 2.5–37.5mm | NR | Electrical | 30 min | Diameter and length: 0.25mm & 40mm Needle brand: Hwato EA: NR | 7 | Once a day, 7 days | NR | NR | NR | NR | Y |
| Chen 2017 | TCM | Y | Y | 9 | ST18, CV17, SI1, ST36, LR3 | 37.5mm | NR | Manual | NR | NR | NR | NR | Y | NR | NR | NR | Y |
| Liu 2017 | TCM | Y | Y | 7 | ST18, CV17, SI1, LR14 | NR | NR | Manual | 25 min | NR | NR | NR | Y | NR | NR | NR | Y |
| Xian 2017 | TCM | Y | Y | 5 | CV17, SI1, ST36 | 10mm | Deqi | Manual | 30 min | NR | 14 | Once a day, 14 days | Y | Y | NR | NR | Y |
| Zheng 2015 | TCM | Y | Y | 5 | CV17, ST36, SI1 | 2.5–50mm | Deqi | Manual | 20 min | Diameter and length: 0.30mm & (25–50) mm Needle brand: Hwato | 7 | Once a day,7 days | NR | Y | NR | NR | Y |

*(Continued)*

**Table 2.** (Continued)

| Study | Acupuncture rational | | | Details of needling | | | | | | | Treatment regimen | | Other components | | Practitioner | Comparator interventions | |
|---|---|---|---|---|---|---|---|---|---|---|---|---|---|---|---|---|---|
| | 1a | 1b | 1c | 2a | 2b | 2c | 2d | 2e | 2f | 2g | 3a | 3b | 4a | 4b | 5 | 6a | 6b |
| Gao 2012 | TCM | Y | Y | 5 | CV17, ST36, BL20 | 12.5–37.5mm | Deqi | Manual | 20 min | Diameter and length: 0.3mm & 25mm Needle brand: NR | 7 | Once a day, 7 days | Y | Y | NR | NR | Y |
| Isabella Neri, 2011 | TCM | Y | Y | 5 | ST18, CV17, SI1 | 10–30mm | Deqi | Manual | 5–30 min | Diameter and length: 0.3mm & 52mm Needle brand: NR | 6 | Twice weekly, 3 weeks | NR | Y | Y | NR | Y |
| Li 2010 | TCM | Y | Y | 5 | ST18, CV17, LR14 | 20–37.5mm | Deqi | Manual | 35 min | Diameter and length: 0.38mm & NR Needle brand: NR | 10 | Once a day, 10 days | NR | Y | NR | NR | Y |
| Zhang 2009 | TCM | Y | Y | 11 | CV17, ST18, ST36, SI1, SI9, PC6 | NR | Deqi | Manual | 30 min | NR | 11 | Once a day, 7 days | Y | Y | NR | NR | Y |
| Zhao 2008 | TCM | Y | Y | 5 | ST18, CV17, SI1 | 6–37.5mm | Deqi | Manual | 30 min | Diameter and length: NR Needle brand: Jiankang | 5 | Once a day, 5 days | NR | Y | NR | NR | Y |
| He 2008 | TCM | Y | Y | 1 | CV17 | 20mm | Deqi | Electrical | 20 min | Diameter and length: 0.35mm & 25mm Needle brand: Ruiqier EA: LH202H apparatus | 3 | Once a day, 3 days | NR | Y | NR | NR | |
| Liu 2006 | TCM | Y | Y | 5 | ST18, CV17, SI1 | NR | NR | Manual | 30 min | NR | 10 | Once a day, 10 days | NR | NR | NR | NR | Y |
| Chen 2000 | TCM | Y | Y | 8 | ST18, CV17, BL20, ST36, RN12 | 7.5–50mm | NR | Manual | 15 min | NR | 5 | once a day, 5 days | Y | NR | NR | NR | Y |
| Zhang 2022 | TCM | Y | Y | 10 | ST36, SP6, BL17, BL23, BL25 | 1.5mm | NR | Manual | 48 h | Diameter and length: 0.2 & 1.5 mm Needle brand: Japanese QingLing | 1 | 3 times/ acupoint | Y | Y | NR | NR | Y |

*Note.* 1a: style of acupuncture, 1b: reasoning for treatment provided, 1c: extent to which treatment was varied, 2a: number of needle insertions per subject per session, 2b: names of points used, 2c: depth of insertion, 2d: response sought, 2e: needle stimulation, 2f: needle retention time, 2g: needle type, 3a: number of treatment sessions, 3b: frequency and duration of treatment sessions, 4a: details of other interventions administered to the acupuncture group, 4b: setting and context of treatment, 5: description of participating acupuncturists, 6a: rationale for the control or comparator, 6b: precise description of the control or comparator, NR: not recorded, Y: yes.

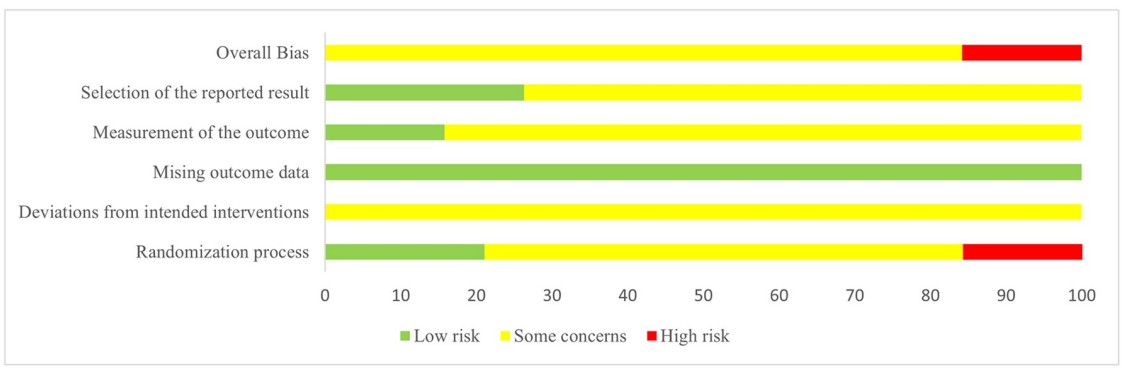

**Fig 2. Risk of bias graph.**

## Meta-analysis

**Serum PRL level.** The serum PRL level was reported in seven studies with a total of 1,055 participants. Meta-analysis results showed that the participants who received acupuncture had higher serum PRL levels compared to those of the control participants (SMD = 1.09, 95% CI: 0.50, 1.68; $P = 0.0003$, $I^2 = 94\%$). Four studies compared acupuncture with CH and showed a significant difference in PRL levels in favor of acupuncture (SMD = 1.03, 95% CI: 0.12, 1.93; $P = 0.03$, $I^2 = 97\%$). Moreover, pooled data from two RCTs indicated that treatment with acupuncture combined with CH was more effective than that with CH alone (SMD = 1.17, 95% CI: 0.38, 1.97; $P = 0.004$, $I^2 = 84\%$). Only one study described serum PRL levels by comparing

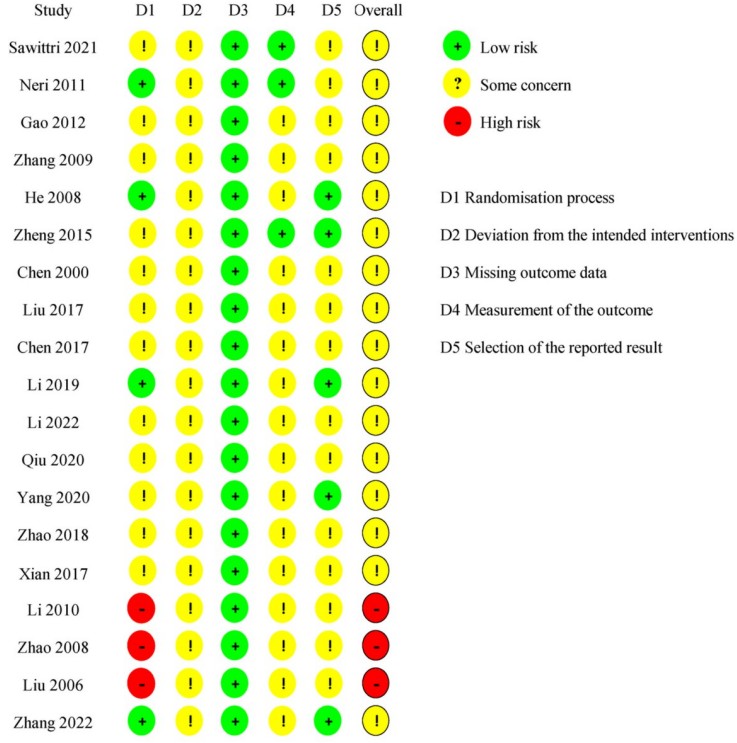

**Fig 3. Risk of bias summary.**

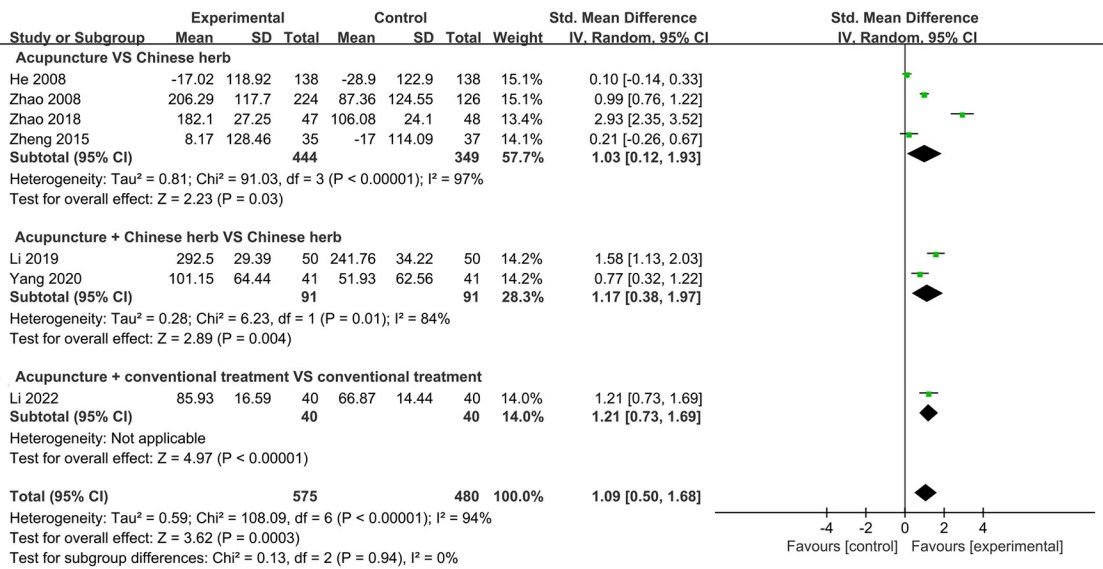

**Fig 4. Forest plot of serum PRL level.**

acupuncture plus CT and CT alone, indicating that the acupuncture group had a better outcome; meta-analysis was not applicable (Fig 4).

Owing to the substantial heterogeneity in our results, we performed a subgroup analysis based on different treatment durations. This analysis revealed that acupuncture treatments with durations of ≤7 days (SMD = 1.03, 95% CI: 0.12, 1.93) and >7 days (SMD = 1.19, 95% CI: 0.72, 1.66) resulted in a statistically significant elevation in PRL levels compared with that in controls. Treatments for a duration >7 days had slightly reduced heterogeneity ($I^2$ = 68%), whereas those with a duration of ≤7 days ($I^2$ = 97%) had higher heterogeneity, indicating that treatment duration may be a source of heterogeneity (S1 Fig).

**MSV.** Five studies with a total of 430 participants measured MSV. The pooled data showed that acupuncture had an overall significant positive effect in increasing MSV (SMD = 1.69 95% CI: 0.53, 2.86; $P$ = 0.004, $I^2$ = 96%). Among these studies, two trials compared acupuncture plus CT with CT alone, and two trials compared acupuncture plus CH with CH alone. The meta-analysis showed that acupuncture plus CT was better than CT alone (SMD = 1.24, 95% CI: 0.40, 2.08; $P$ = 0.004, $I^2$ = 84%). Furthermore, treatment with acupuncture plus CH was more effective in increasing MSV than that with CH alone (SMD = 3.06, 95% CI: 2.09, 4.04; $P$ < 0.00001, $I^2$ = 79%). Only one study compared the efficacy of acupuncture with that of CT; therefore, a meta-analysis was not performed, and the results suggested that acupuncture was not as beneficial as CH for women with PH (Fig 5).

Given the high heterogeneity of the pooled results, subgroup analyses were performed according to the different treatment durations. Results showed that acupuncture increased MSV in patients with a treatment duration >7 days (SMD = 2.58, 95% CI: 1.52, 3.65) but not in those with a treatment duration of ≤7 days (SMD = 0.35, 95% CI: –0.51, 1.21). Treatment durations of less than ($I^2$ = 83%) and greater than ($I^2$ = 91%) 7 days did not show a dramatic decline in heterogeneity (S1 Fig).

**TER.** Thirteen studies involving a total of 1,637 participants reported the effects of acupuncture on TER. The meta-analysis indicated that acupuncture significantly improved the TER compared to the control (RR = 1.25, 95% CI: 1.10; 1.42, $P$ = 0.0008, $I^2$ = 93%). Pooled

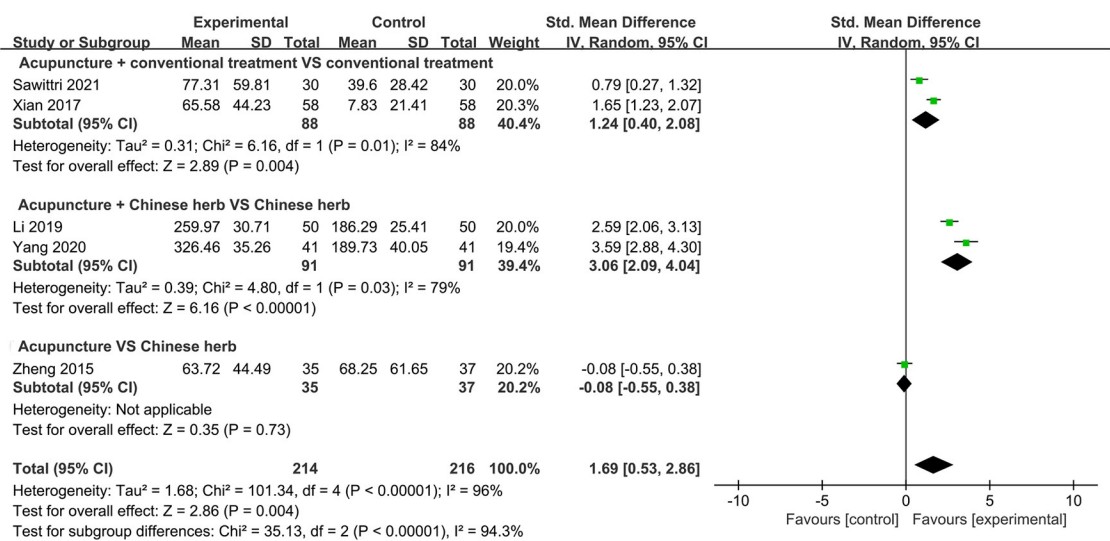

**Fig 5. Forest plot of MSV.**

analysis of seven RCTs showed that acupuncture plus CH treatment was more effective in improving the TER than that with CH treatment alone (RR = 1.24, 95% CI: 1.06, 1.45; $P = 0.007$, $I^2 = 88\%$). In contrast, the merged data from five studies showed that acupuncture was not superior to CH (RR = 1.07, 95% CI: 0.97, 1.19; $P = 0.19$, $I^2 = 75\%$). Only one study compared acupuncture versus CT and acupuncture plus CT versus CT alone; therefore, a meta-analysis could not be performed. In the trials, both acupuncture plus CT and acupuncture alone significantly improved the TER compared to that by CT alone (Fig 6).

The subgroup analysis based on different treatment durations showed that acupuncture increased the TER in patients with treatment durations >7 days (RR = 1.43, 95% CI: 0.97, 2.10) and ≤7 days (RR = 1.11, 95% CI: 1.01, 1.22) more than that in controls. No subgroup showed substantially reduced heterogeneity ($I^2 = 77\%$; $I^2 = 95\%$) (S1 Fig).

**MFD.** Four RCTs with a total of 587 participants reported data regarding MFD. Pooled results revealed that, overall, no significant difference was observed between the acupuncture and control groups (SMD = 1.17, 95% CI: –0.09, 2.42; $P = 0.07$, $I^2 = 98\%$). Furthermore, no significant difference was observed between the acupuncture and CH groups (SMD = 0.32, 95% CI: –0.67, 1.32; $P = 0.52$, $I^2 = 94\%$). Only one trial investigated the effects of acupuncture plus CH versus CH alone and acupuncture plus CT versus CT alone; therefore, a quantitative synthesis could not be performed. The trials indicated that acupuncture plus either CH or CT treatment improved MFD to a greater degree than that with CH or CT alone (Fig 7).

Subgroup analysis based on different treatment durations indicated that neither acupuncture therapy for ≤7 days (SMD = 0.32, 95% CI: –0.67, 1.32) nor for >7 days (SMD = 2.02, 95% CI: –0.45, 4.49) had an advantage in improving MFD compared to that in controls. No subgroup showed significantly reduced heterogeneity ($I^2 = 94\%$; $I^2 = 89\%$) (S1 Fig).

**EBR.** Two trials with a total of 164 participants explored the effects of acupuncture on EBR. The pooled results showed that participants who received acupuncture had a significantly greater improvement in EBR compared to that in controls (RR = 2.01, 95% CI: 1.07, 3.78; $P = 0.03$, $I^2 = 69\%$) (Fig 8). One study compared acupuncture with CT, and another compared acupuncture plus CT with CT alone; therefore, a meta-analysis was not feasible. These studies

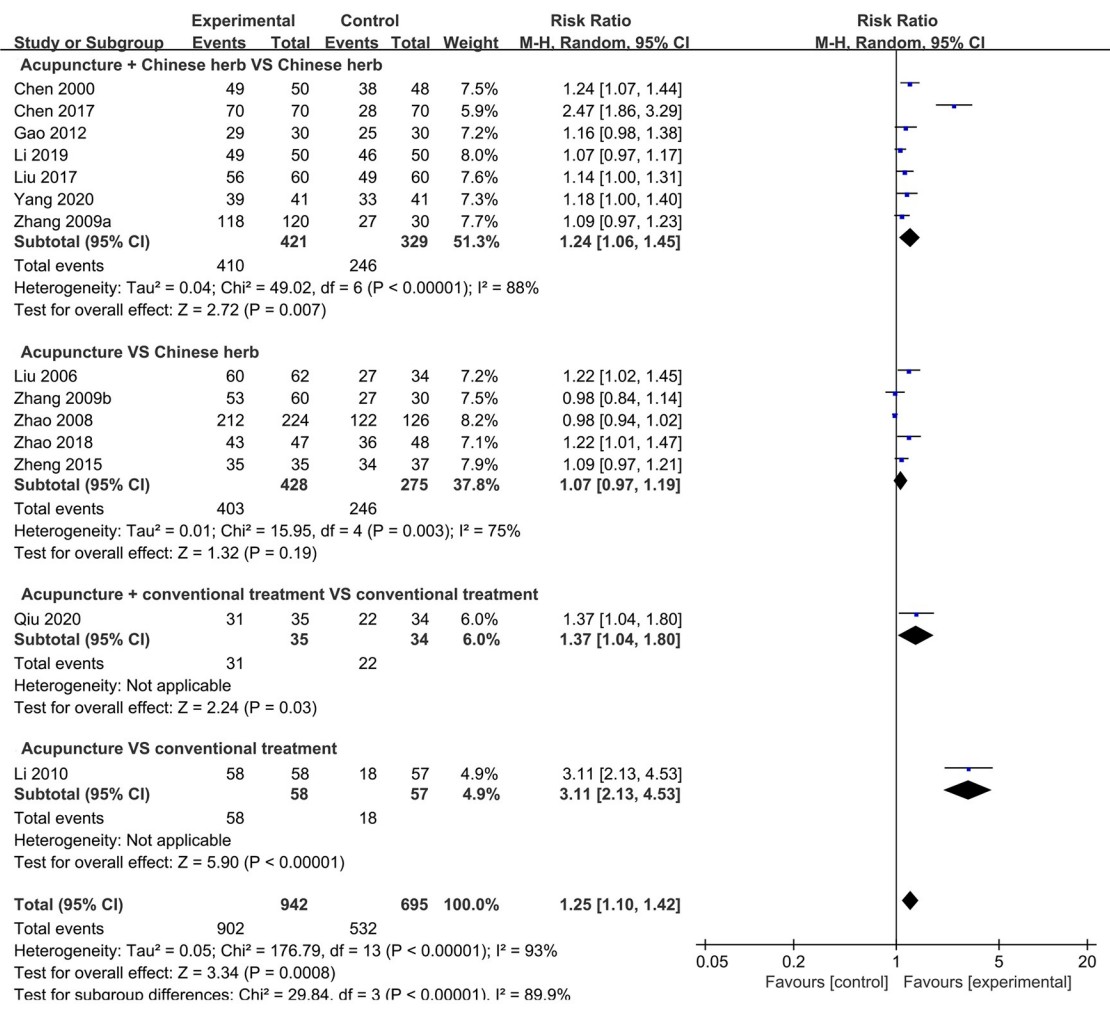

**Fig 6. Forest plot of TER.**

suggested that both the acupuncture and acupuncture plus CT groups showed a greater increase in EBR than that by CT alone.

**Adverse events.** Four studies [32, 38, 44, 48] examined the adverse events associated with the included treatments. Three recorded minor adverse events due to acupuncture, such as skin itching, dizziness, and negative sensations, which resolved after treatment. One reported no complications in any of the studied patients (Table 3); no severe adverse events were reported.

## Sensitivity analysis

Sensitivity analyses were performed based on the outcome of serum PRL level and TER. For serum PRL level, there was no evident change in the pooled effect size when comparing acupuncture and CH after excluding each study individually. For TER, in comparisons of acupuncture versus CH and acupuncture plus CH versus CH alone, the pooled effect size did not change significantly when studies were eliminated individually (S2 Fig); however, the heterogeneity greatly decreased ($I^2$ = 0%) when comparing acupuncture plus CH versus CH alone

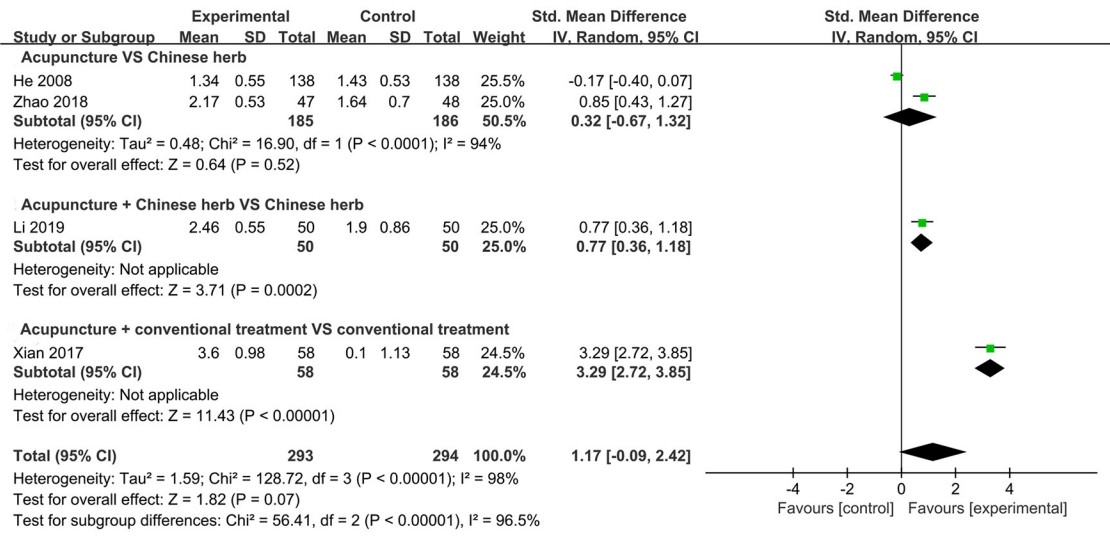

**Fig 7. Forest plot of MFD.**

when the study by Chen et al. [43] was removed (S3 Fig). Sensitivity analyses were not performed for the other outcomes and comparisons, as very few studies remained.

## Publication bias

The publication bias was assessed using funnel plots and Egger's test. Regarding the funnel plot of TER (Fig 9), most included studies were asymmetrically distributed on the two sides of the midline, demonstrating the presence of a reporting bias. However, funnel plots were not used for other outcomes because the number of included trials did not exceed 10.

Concerning Egger's test (S2 Table), the serum PRL level ($P = 0.186$), MSV ($P = 0.269$), and MFD ($P = 0.065$) had no publication bias. Conversely, significant publication biases were detected for TER ($P = 0.000$). Egger's test was not applied to EBR due to the limited number of included studies.

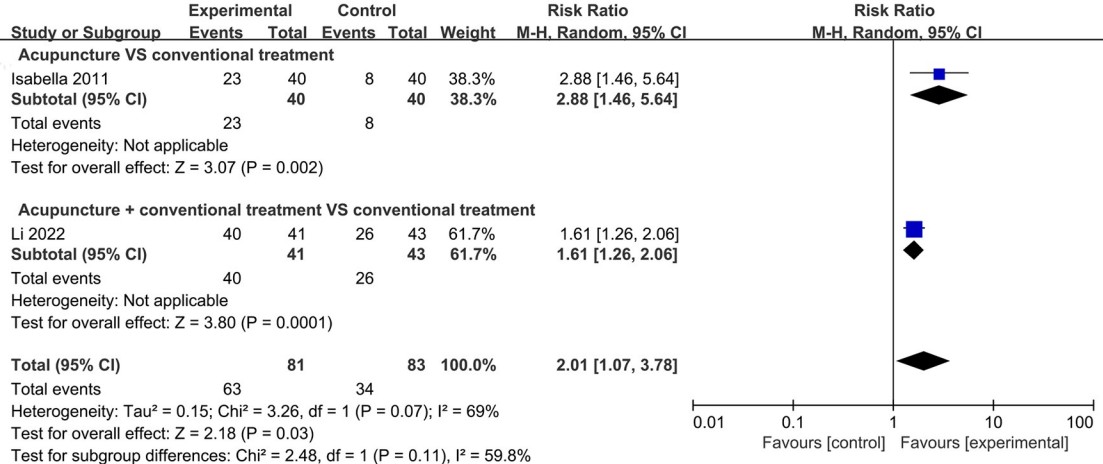

**Fig 8. Forest plot of EBR.**

**Table 3. Adverse events in included RCTs.**

| Interventions | Study | Number of adverse events | Details of adverse events |
|---|---|---|---|
| MA | Zheng, 2015 [44] | 6 | 3 cases of skin itching, 3 cases of dizziness. |
| MA | Isabella Neri, 2011 [38] | 3 | 3 cases of having had any negative sensations, such as fear of needle. |
| MA + CH | Liu, 2017 [48] | 8 | 3 cases of loss of appetite, 5 cases of nausea, 3 cases of breast distension. |
| CH | Liu, 2017 [48] | 14 | 6 cases of loss of appetite, 5 cases of nausea, 3 cases of breast distension. |
| CH | Zheng, 2015 [44] | 9 | 1 case of skin itching, 2 cases of dizzy, 6 cases of gastrointestinal discomfort. |

*Notes.* MA: manual acupuncture, CH: Chinese herb.

## Quality of evidence

The quality of evidence for the five outcomes (improvement in serum PRL level, MSV, TER, MFD, and EBR) was measured using the GRADE tool. All outcomes were rated as critically low. Low methodological quality, inconsistencies, and publication bias were the main reasons for the low grades. The details are presented in Table 4.

## Discussion

Although the benefits of breastfeeding for mothers and babies are widely recognized, breast milk insufficiency remains a major barrier to breastfeeding. During lactation, interventions for

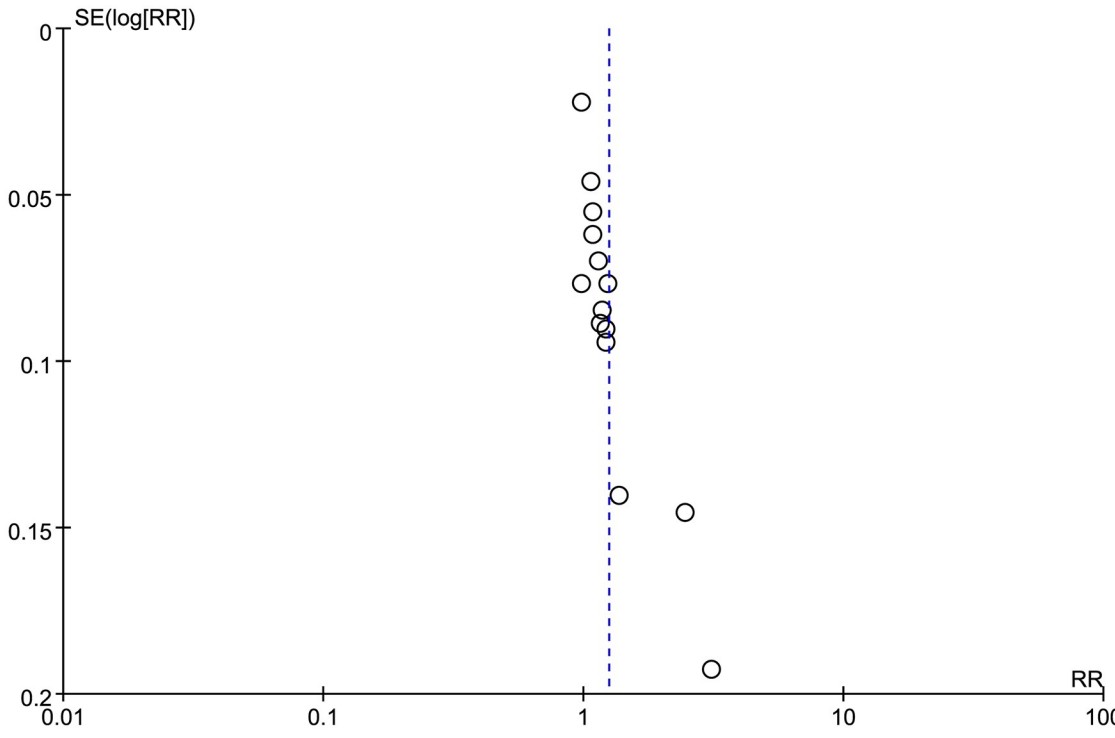

**Fig 9. Funnel plot of improvement in TER.**

**Table 4. GRADE quality of evidence for the outcomes included in the study.**

| Outcomes | Included studies (participants) | RoB | Inconsistency | Indirectness | Imprecision | Publication bias | Large effect | Dose-response | All plausible confounders | Quality |
|---|---|---|---|---|---|---|---|---|---|---|
| Serum PRL level | 7 (1055) | Serious[a] | Serious[b] | Not serious | Not serious | Strongly suspected[c] | No | No | No | Critically low |
| MSV | 5 (430) | Serious[a] | Serious[b] | Not serious | Not serious | Strongly suspected[c] | No | No | No | Critically low |
| TER | 14 (1697) | Serious[a] | Serious[b] | Not serious | Not serious | Strongly suspected[c] | No | No | No | Critically low |
| MFD | 4 (587) | Serious[a] | Serious[b] | Not serious | Not serious | Strongly suspected[c] | No | No | No | Critically low |
| EBR | 2 (164) | Serious[a] | Serious[b] | Not serious | Not serious | Strongly suspected[c] | No | No | No | Critically low |

*Notes*. GRADE: Grading of Recommendations Assessment, Development and Evaluation, RoB: risk of bias, PRL: prolactin, MSV: milk secretion volume, TER: total effective rate, MFD: mammary fullness degree, EBR: exclusive breastfeeding rate.

[a]Most of the included studies have risk of bias.

[b]Substantial meta-analysis heterogeneity.

[c]Potential publication bias.

managing PH should be carried out with caution. The Academy of Breastfeeding Medicine does not recommend any medications for PH [16], and several women with PH are unwilling to take medications because of concerns about potential harm to their infants. Thus, alternative and complementary PH therapies, including acupuncture, have received increasing attention. Previous studies have reported that acupuncture may improve breast milk production [32, 39]; however, high-quality evidence supporting this outcome remains scarce. This review presents a comprehensive assessment of the current evidence on acupuncture efficacy in managing PH.

## Summary of main results

This systematic review included 19 studies involving 2,400 women with PH during lactation. Overall, the results of the meta-analysis suggested that acupuncture can significantly increase serum PRL level, MSV, TER, and EBR in patients with PH, indicating that acupuncture is an effective method to improve milk production in women with PH. However, participants receiving acupuncture showed no significant improvement in MFD compared to that in controls, possibly because of the limited studies included. In addition, we found that acupuncture combined with CH or CT was more effective than CH or CT alone in increasing serum PRL level, MSV, and TER, indicating superior efficacy in treating PH. When compared with CH, acupuncture showed significantly higher serum PRL levels; nevertheless, no difference was observed in TER and MFD, suggesting that acupuncture and CH may have similar effects in boosting breast milk production. Subgroup analysis based on treatment durations demonstrated that acupuncture treatment for >7 days and ≤7 days significantly increased PRL levels and TER; however, only treatments exceeding 7 days surpassed the controls in augmenting MSV. This is likely because breast milk production only increases when the cumulative effect of acupuncture is reached. Concerning safety, minor adverse effects due to acupuncture were recorded in three studies, which resolved after treatment; therefore, acupuncture can be considered a relatively safe intervention for women with PH. However, the included RCTs had moderate or low methodological quality, and the quality of evidence for the outcomes derived from the RCTs was critically low; thus, these conclusions should be interpreted with caution.

**Explanation of intervention mechanisms.** Lactation, the secretion of milk from the mammary gland, is affected by a complex hormonal network. Among the most important hormones are PRL and oxytocin (OT), which are vital in lactogenesis [58]. PRL is essential in promoting mammary gland differentiation, preparing the breast for lactation, and stimulating milk protein and lactose synthesis [59]. PRL levels substantially increase during pregnancy and remain elevated after delivery. As breastfeeding intensity decreases gradually, basal PRL levels return to normal gradually. PRL deficiency can cause lactation insufficiency [60]. Pharmacological galactagogues function primarily by stimulating PRL secretion [21]. The meta-analyses demonstrated that acupuncture significantly increased serum PRL levels in patients with PH, thereby promoting PRL release. Given the essential role of PRL in regulating milk secretion, it may be a potential target for acupuncture in influencing breast milk production. Moreover, intermittent pulsatile OT secretion is necessary for the milk ejection reflex during lactation, and dysregulation in OT secretion can cause maternal hypogalactia [61]. However, none of the included studies measured serum OT levels in patients with PH.

Additionally, various factors contribute to PH, with different pathologies. Acupuncture may mediate complex processes involving interactions between physical and physiological factors [62]. Acupuncture seems effective in treating postpartum depression [63], which may be a contributing factor to PH [64]. However, no study has measured the emotional status of women with PH.

## Implications for clinical practice and further research

The results of this study may provide guidance for the clinical treatment of PH. First, acupuncture therapy may be an effective option for improving breast milk production in women with PH. Additionally, healthcare workers may find that acupuncture combined with CH or CT lasting over 7 days is a suitable approach for managing PH. Second, acupoint selection affects the efficacy of acupuncture, with CV17, SI1, ST18, and ST36 being the most commonly used acupoints for PH treatment, consistent with clinically recommended primary acupoints [65]. Third, accurate TCM syndrome differentiation can guide treatment [55]. Women with PH with different syndrome types should be provided with tailored acupuncture prescriptions. In addition, considering that a few participants in the acupuncture groups experienced skin itching and negative sensations, this treatment may not be suitable for those who fear needling or are allergic to metals.

Given the poor methodological quality, high heterogeneity, and limited strength of evidence of previous studies, more trials with rigorous study design and methodology are warranted. Appropriate randomization methods, blinding, allocation concealment, and intention-to-treat analyses should be used to reduce the risk of bias. Additionally, researchers should register their studies in a clinical trial registry before recruiting participants, formulate strict diagnostic inclusion and exclusion criteria for PH, adopt unified efficacy evaluations and standard treatment plans, and conduct and strictly report the research following the STRICTA and PRISMA guidelines. Moreover, some critical outcome indicators, such as the onset of breastfeeding (time, amount), serum OT levels, and emotional status (depression, anxiety), should be included in future studies. Similarly, key variables such as specific manipulations, optimal acupoints, intervention timings, and effective frequencies of acupuncture in promoting breast milk production should also be investigated.

## Strengths and limitations

To our knowledge, this is the first systematic review to examine the efficacy and safety of acupuncture for breast milk production in PH. The protocol was published before implementing

the study. Moreover, STRICTA was used to assess the details of the acupuncture studies, and RoB 2.0 and GRADE were used to evaluate the methodological and evidence qualities of the RCTs. This ensured more reliable evidence that can be applied in clinical management and referenced in future research. Furthermore, this systematic review has been reported following the PRISMA reporting guidelines.

Nevertheless, this study had some limitations. First, most of the included studies had methodological flaws, which contributed to poor quality studies being included in this analysis. These were assessed with some concern, mainly owing to a lack of or weak descriptions of blinding, randomization methods, allocation concealment, and missing information. These factors introduce performance, selection, and detection biases. Second, substantial heterogeneity in serum PRL levels, MSV, TEF, and MFD were observed. One explanation is that PRL levels are influenced by several factors [66], and no particular cut-off value for PRL is associated with adequate milk production [23]. The outcomes MSV, TER, and MFD were measured in multiple ways and can be easily affected by the clinicians' experience. Other factors, such as variation in infants' ages and the diversity of acupuncture interventions, can also partially explain the heterogeneity in the results. Third, publication bias was detected in TER as a secondary outcome measure; only two of the studies included in this review were not conducted in China, which might have led to reporting bias. The above limitations are the main causes for this review's "critically low" level of evidence certainty.

## Conclusions

This systematic review and meta-analysis demonstrated that acupuncture may be an effective and safe approach for treating PH. However, the quality of the evidence remains critically low, and more high-quality clinical trials with standardized protocols are needed for further validation.

## Supporting information

**S1 Fig. Subgroup analyses of the outcomes of the included studies.**
(DOCX)

**S2 Fig. Sensitivity analyses of different outcomes.**
(DOCX)

**S3 Fig. Forest plot of TER for acupuncture + Chinese herbs vs. Chinese herbs with the study by Chen et al. (2017) removed.**
(DOCX)

**S1 Table. Excluded full-text articles with reasons for exclusion.**
(DOCX)

**S2 Table. Egger's test for different outcomes.**
(DOCX)

**S1 File. PRISMA checklist.**
(DOCX)

**S2 File. Search strategies for each database.**
(DOCX)

## Acknowledgments

The authors appreciate all the reviewers for their assistance.

## Author Contributions

**Conceptualization:** Qiong-Nan Bao.

**Data curation:** Yuan-Fang Zhou, Ya-Qin Li, Shao-Jun Xu.

**Formal analysis:** Man-Ze Xia, Zheng-Hong Chen, Wan-Qi Zhong.

**Funding acquisition:** Fan-Rong Liang.

**Methodology:** Ya-Qin Li, Xin-Yue Zhang.

**Supervision:** Fan-Rong Liang.

**Validation:** Zhen-Yong Zhang.

**Visualization:** Jin Yao, Ke-Xin Wu.

**Writing – original draft:** Qiong-Nan Bao.

**Writing – review & editing:** Zi-Han Yin.

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
