## [Decision Letter · Decision Letter 0]

17 Oct 2023

PONE-D-23-25490Efficacy and safety of acupuncture for postpartum hypogalactia: a systematic review and meta-analysis of randomized controlled trialsPLOS ONE

Dear Dr. Liang

Thank you for submitting your manuscript to PLOS ONE. After careful consideration, we feel that it has merit but does not fully meet PLOS ONE’s publication criteria as it currently stands. Therefore, we invite you to submit a revised version of the manuscript that addresses the points raised during the review process.

We look forward to receiving your revised manuscript.

Kind regards,

Benjamin Jun Jie Seng, MD

Academic Editor

PLOS ONE

Additional Editor Comments:

Dear Professor Liang,

Thank you for the submission to PLOS One.

Below are the reviewers' comments for your consideration.

Methods

- The search period is slightly outdated and nearly done 1 year from date of submission

-> It will benefit from an update

Table 2

- Standardize the reporting and number of significant figures reported for the results

- Suggest to put the actual treatment instituted in the control group

-> This has implications on the suitaibility of meta-analyses as if there are drastically different control measures implemented, the pooling of the results may not be appropriate.

-> the authors may need to check on this

Discussion

- Brief discussion should be made with regards to role of acupuncture in postpartum hypogalactia as to how it should be used in current practice

- Are there any patients whom acupuncture should not be used?

Reviewers' comments:

Reviewer's Responses to Questions

**Comments to the Author**

1. Is the manuscript technically sound, and do the data support the conclusions?

Reviewer #1: Yes

Reviewer #2: Yes

2. Has the statistical analysis been performed appropriately and rigorously

Reviewer #1: No

Reviewer #2: Yes

3. Have the authors made all data underlying the findings in their manuscript fully available?

Reviewer #1: Yes

Reviewer #2: Yes

4. Is the manuscript presented in an intelligible fashion and written in standard English?

Reviewer #1: No

Reviewer #2: Yes

5. Review Comments to the Author

Reviewer #1: Were there any attempts made to look for unpublished data which could have affected the evaluation of the meta-analysis?

Table 2: what is TCM syndrome differentiation? Table 2 is also difficult to read with many acronyms and many different outcomes being reviewed. Are there certain standardized or commonly used parameters we can focus on instead of evaluating such a wide range of efficacy and safety parameters?

Table 5: Inappetence > can be replaced with "Loss of appetite"

"Dizzy" > can be replaced with "Dizziness"

"Breast distending" > can be replaced with "breast distension"

Please complete what "3 cases of gastrointestinal" means. Do you mean "gastrointestinal discomfort"?

Reviewer #2: Interesting study. Some issues need clarifications.

1 Write the full details of search strategy with the keywords used and results of each data base individually

2. Study characteristics add the inclusion and exclusion criteria of participants

3. Add the registration details of each study prospective or retrospective or not registered

4. Subgroup analysis according to time frequency type and duration of intervention should be added

6. PLOS authors have the option to publish the peer review history of their article (what does this mean?). If published, this will include your full peer review and any attached files.

Reviewer #1: No

Reviewer #2: **Yes: **Ahmed M Maged

---

## [Author Response · Author response to Decision Letter 0]

18 Nov 2023

Dear Editors and Reviewers:

We sincerely appreciate for the editors and reviewers’ comments concerning our manuscript entitled “Efficacy and safety of acupuncture for postpartum hypogalactia: A systematic review and meta-analysis of randomized controlled trials”. Those comments are all valuable and very helpful for revising and improving our paper, as well as the important guiding significance to our review. We have studied comments carefully and have made correction which we hope meet PLOS ONE’s publication criteria. Revised portions are marked red in the paper. The main corrections in the paper and the responds to the editor and reviewers’ comments are as following:

Editor’s Comments: 

MD. Benjamin Jun Jie Seng:

1. Please ensure that your manuscript meets PLOS ONE's style requirements, including those for file naming. The PLOS ONE style templates can be found at.

Response: Many thanks for your kindly comment. We have carefully read the PLOS ONE style templates you provided, and revised our manuscript according to the requirements, including file naming, title, authors, affiliations and main body.

Response: Thanks for your significant comment. We have checked and revised the grant information in the ‘Funding Information’ section. This study was financially supported by the “Central Financial Transfer Payment to Local Projects in 2022 of National Administration of Traditional Chinese Medicine”, which does not have a specific grant number. Fanrong Liang was the study funder. 

Response: Thanks for this important guidance. I have seen the instructed video and linked an ORCID iD in my Editorial Manager account. 

Additional Editor Comments:

1. Methods

- The search period is slightly outdated and nearly done 1 year from date of submission

-> It will benefit from an update

Response: Thanks for your significant comment. We have performed an updated search on October 19, 2023. 1 more RCT was included in this systematic review and relevant results have been updated. 

2. Table 2

- Standardize the reporting and number of significant figures reported for the results

- Suggest to put the actual treatment instituted in the control group

-> This has implications on the suitability of meta-analyses as if there are drastically different control measures implemented, the pooling of the results may not be appropriate.

-> the authors may need to check on this

Response: Thanks a lot for your significant suggestion. We have standardized the reporting and number of significant figures reported for the results. Based on currently available original data, sham acupuncture, conventional treatment (breast sucking, postpartum routine care, breastfeeding education) and Chinese herb were included as control in this systematic review. Studies used interventions such as drugs, galactagogues, massage, or any other complementary therapy in control group were not included. The pooled data from included RCTs are divided into comparators including acupuncture + control vs. control and acupuncture vs. control. We have described the detailed control measures and treatment in the Methods and Results section.

3. Discussion

- Brief discussion should be made with regards to role of acupuncture in postpartum hypogalactia as to how it should be used in current practice

- Are there any patients whom acupuncture should not be used?

Response: Thank you for this precious suggestion. Our findings may provide guidance for the clinical treatment of PH. We have discussed how to use acupuncture to treat postpartum hypogalactia in practice from the perspective of better combination modality, treatment duration, acupoint selection, and traditional Chinese medicine (TCM) syndrome differentiation. Concerning safety, minor adverse effects of acupuncture were recorded in three studies, and these responses returned to normal after treatment. Therefore, acupuncture could be considered a relatively safe intervention to improve milk supply for women with PH. Considering that few participants in the acupuncture group experienced skin itching and negative sensations, it may not suitable for those who fear of needling or allergic to metal. We have added the in the Discussion section.

Reviewer #1:

1. Were there any attempts made to look for unpublished data which could have affected the evaluation of the meta-analysis?

Response: Thanks a lot for your important comment. We have retrieved clinical registry platform for unpublished data, and no available unpublished data was found.

2. Table 2: what is TCM syndrome differentiation? Table 2 is also difficult to read with many acronyms and many different outcomes being reviewed. Are there certain standardized or commonly used parameters we can focus on instead of evaluating such a wide range of efficacy and safety parameters?

Response: Thanks for your significant suggestion. Treatment based on TCM syndrome differentiation is the core principle of TCM; that is means taking the individual as the study’s starting point and analyzing the pathophysiological characteristics, nature of lesions and development trend according to their clinical signs and symptoms, so as to develop the corresponding TCM treatments. Many studies have demonstrated that TCM syndrome differentiation have value in predicting prognosis and improving intervention effects in various diseases. In this systematic review, thirteen of the included studies included participants based on the TCM syndrome differentiation, including deficiency of Qi and blood syndrome; liver-Qi stagnation syndrome; deficiency of spleen-Qi syndrome; phlegm dampness and blood stasis syndrome; Qi deficiency and blood stasis syndrome. We have added information regarding Traditional Chinese medicine (TCM) syndrome differentiation in study characteristic.

We have revised the Table 2. It was complicated and hard to understand. For outcome measures, we retained some standardized and commonly used parameters, including serum prolactin level, milk secretion volume, mammary fullness degree, total effective rate, exclusive breastfeeding rate, and adverse events. 

3. Table 5: Inappetence > can be replaced with "Loss of appetite"

"Dizzy" > can be replaced with "Dizziness"

"Breast distending" > can be replaced with "breast distension"

Please complete what "3 cases of gastrointestinal" means. Do you mean "gastrointestinal discomfort"?

Response: Thank you very much for your careful review. We have modified the above inappropriate words in the Table 3. 

Reviewer #2: 

Interesting study. Some issues need clarifications.

1. Write the full details of search strategy with the keywords used and results of each data base individually

Response: Many thanks for your significant comment. We have added the detailed search strategies of each database in the S1 file.

2. Study characteristics add the inclusion and exclusion criteria of participants

Response: Thanks for your important comment. We have added information regarding the inclusion and exclusion criteria of participants in the Study characteristics section. A total of eleven of the included studies described the inclusion and exclusion criteria of participants in detail.

3. Add the registration details of each study prospective or retrospective or not registered

Response: Thanks for your kindly recommendation. All the included studies in this systematic review have not registered in the clinical registry platform in advance. We have added related information in the Results and Discussion section.

4. Subgroup analysis according to time frequency type and duration of intervention should be added

Response: Thanks a lot for your advice. We have added subgroup analyses according to duration of intervention, the results have been presented in the Results section. But subgroup analysis based on time frequency type cannot be performed, because all studies except 1 were treated once a day.

---

## [Decision Letter · Decision Letter 1]

18 Dec 2023

PONE-D-23-25490R1Efficacy and safety of acupuncture for postpartum hypogalactia: A systematic review and meta-analysis of randomized controlled trialsPLOS ONE

Dear Dr. Liang,

Thank you for submitting your manuscript to PLOS ONE. After careful consideration, we feel that it has merit but does not fully meet PLOS ONE’s publication criteria as it currently stands. Therefore, we invite you to submit a revised version of the manuscript that addresses the points raised during the review process.

We look forward to receiving your revised manuscript.

Kind regards,

Benjamin Jun Jie Seng, MD

Academic Editor

PLOS ONE

Additional Editor Comments:

Dear authors

Pls refer to comments by the reviewers and provide your point by point reply. Thank you

Reviewers' comments:

Reviewer's Responses to Questions

**Comments to the Author**

1. If the authors have adequately addressed your comments raised in a previous round of review and you feel that this manuscript is now acceptable for publication, you may indicate that here to bypass the “Comments to the Author” section, enter your conflict of interest statement in the “Confidential to Editor” section, and submit your "Accept" recommendation.

Reviewer #1: All comments have been addressed

Reviewer #3: (No Response)

2. Is the manuscript technically sound, and do the data support the conclusions?

Reviewer #1: Yes

Reviewer #3: Partly

3. Has the statistical analysis been performed appropriately and rigorously? 

Reviewer #1: Yes

Reviewer #3: No

4. Have the authors made all data underlying the findings in their manuscript fully available?

Reviewer #1: Yes

Reviewer #3: Yes

5. Is the manuscript presented in an intelligible fashion and written in standard English?

Reviewer #1: Yes

Reviewer #3: Yes

6. Review Comments to the Author

Reviewer #1: No further comments, my comments have been addressed.

This topic is of interest as PH is a prevalent issue among breastfeeding mothers. It is useful to know the utility of accupuncture in this area as this is something that is not commonly considered by layman as treatment for PH

Reviewer #3: 1. The studies differ significantly in terms of interventions and control arms. Authors should present data as following comparators (MA+CT vs CT, MA+CH vs CH, MA vs CH and MA vs CT, etc) in one forest plot as subgroup or refrain from conducting meta-analysis and present finding as systematic review only. This will reduce the number of figures and give better clarity to manuscript.

2. Other concern is inclusion of selection of studies that did not report the selection criteria (8 studies) and treatment duration (3 studies) besides high concern in randomization. The findings of meta-analysis depend on kind of included studies. Overall, poor quality studies are included in the meta-analysis.

3. I do not agree with GRADE evidence quality of outcome. Authors has considered publication bias “undetected” for the outcomes “PRL, MSV, MFD and EBR”. However, authors have made no attempt to detect publication bias for these outcomes due to less than 10 studies. I do not agree with this assessment. Funnel plots are most likely asymmetrical for these outcomes. Publication should be “suspected” for these outcomes. The quality of evidence must be downgraded to “Very low/critically low”.

7. PLOS authors have the option to publish the peer review history of their article (what does this mean?). If published, this will include your full peer review and any attached files.

Reviewer #1: No

Reviewer #3: **Yes: **Tejas K. Patel

---

## [Author Response · Author response to Decision Letter 1]

16 Feb 2024

Dear Editors and Reviewers:

We sincerely appreciate the editors and reviewers’ comments concerning our manuscript entitled “Efficacy and safety of acupuncture for postpartum hypogalactia: A systematic review and meta-analysis of randomized controlled trials”. Those comments are all valuable and very helpful for revising and improving our paper, as well as the important guiding significance to our review. We have studied the comments carefully and have made corrections to meet PLOS ONE’s publication criteria. The revised portions are marked red in the paper. The main corrections in the paper and the response to the editor and reviewers’ comments are as follow:

Reviewers' comments:

Reviewer's Responses to Questions

Comments to the Author

1. If the authors have adequately addressed your comments raised in a previous round of review and you feel that this manuscript is now acceptable for publication, you may indicate that here to bypass the “Comments to the Author” section, enter your conflict of interest statement in the “Confidential to Editor” section, and submit your "Accept" recommendation.

Reviewer #1: All comments have been addressed

Response: We greatly appreciate your review and support for the study.

Reviewer #3: (No Response)

Response: We are grateful for your significant comments. We have carefully read and addressed the manuscript in this round according to your comments.

2. Is the manuscript technically sound, and do the data support the conclusions?

Reviewer #1: Yes

Response: Thank you very much for your agreement.

Reviewer #3: Partly

Response: We extend our many thanks for your review. We have further modified the results of the meta-analysis and presented it.

3. Has the statistical analysis been performed appropriately and rigorously?

Reviewer #1: Yes

Response: Thank you very much for this approval.

Reviewer #3: No

Response: We sincerely thank you for your comments. We have revised the statistical analysis in a more appropriate manner.

4. Have the authors made all data underlying the findings in their manuscript fully available?

Reviewer #1: Yes

Response: Many thanks for your consent.

Reviewer #3: Yes

Response: Many thanks for your consent.

5. Is the manuscript presented in an intelligible fashion and written in standard English?

Reviewer #1: Yes

Response: Thank you very much for this ratification.

Reviewer #3: Yes

Response: Thank you very much for this ratification.

6. Review Comments to the Author

Reviewer #1: No further comments, my comments have been addressed.

This topic is of interest as PH is a prevalent issue among breastfeeding mothers. It is useful to know the utility of acupuncture in this area as this is something that is not commonly considered by layman as treatment for PH.

Response: Thank you very much for your acceptance and approval.

Reviewer #3: 1. The studies differ significantly in terms of interventions and control arms. Authors should present data as following comparators (MA+CT vs CT, MA+CH vs CH, MA vs CH and MA vs CT, etc) in one forest plot as subgroup or refrain from conducting meta-analysis and present finding as systematic review only. This will reduce the number of figures and give better clarity to manuscript.

Response: Thank you for your comments. We have modified the meta-analysis. For each outcome, different comparators (MA+CT vs CT, MA+CH vs CH, MA vs CH and MA vs CT, etc) were presented as subgroups in one forest plot. The number of figures has been reduced, and all results of meta-analysis have been corrected.

2. Other concern is inclusion of selection of studies that did not report the selection criteria (8 studies) and treatment duration (3 studies) besides high concern in randomization. The findings of meta-analysis depend on kind of included studies. Overall, poor quality studies are included in the meta-analysis.

Response: Thank you for your comments. The studies that did not report the selection criteria or treatment duration have been rated as “some concern” in the domain of selection of reported results. We conducted a systematic search through databases from their establishment to October 2023; unfortunately, this led to studies of poor quality being included in this systematic review and meta-analysis, which is a significant limitation of this study. We have described this limitation and put forward relevant recommendation for future research in the Discussion section.

3. I do not agree with GRADE evidence quality of outcome. Authors has considered publication bias “undetected” for the outcomes “PRL, MSV, MFD and EBR”. However, authors have made no attempt to detect publication bias for these outcomes due to less than 10 studies. I do not agree with this assessment. Funnel plots are most likely asymmetrical for these outcomes. Publication should be “suspected” for these outcomes. The quality of evidence must be downgraded to “Very low/critically low”.

Response: Many thanks for your important comments. We have revised the assessment of publication bias into “strongly suspected” for the outcomes of serum prolactin level, milk secretion volume, mammary fullness degree, and exclusive breastfeeding rate, and the quality of evidence has been downgraded to “critically low”.

---

## [Editor Report · Decision Letter 2]

27 Mar 2024

PONE-D-23-25490R2Efficacy and safety of acupuncture for postpartum hypogalactia: A systematic review and meta-analysis of randomized controlled trialsPLOS ONE

Dear Dr. Liang,

Thank you for submitting your manuscript to PLOS ONE. After careful consideration, we feel that it has merit but does not fully meet PLOS ONE’s publication criteria as it currently stands. Therefore, we invite you to submit a revised version of the manuscript that addresses the points raised during the review process.

Multiple grammatical errors still exist. Suggest to add in the databases searched for this review We look forward to receiving your revised manuscript.

Kind regards,

Benjamin Jun Jie Seng, MD

Academic Editor

PLOS ONE

Journal Requirements:

Additional Editor Comments:

Dear authors,

The revision are mostly satisfactory.

Minor comments

- Multiple grammatical errors still remains

- Dear authors,

Abstract

- The databases used for the review should be stipulated

---

## [Author Response · Author response to Decision Letter 2]

30 Apr 2024

Dear Editors:

We sincerely appreciate for the journal requirements and editor's comments concerning our manuscript entitled “Efficacy and safety of acupuncture for postpartum hypogalactia: A systematic review and meta-analysis of randomized controlled trials”. Those comments are all valuable and very helpful for revising and improving our paper, as well as the important guiding significance to our review. We have studied comments carefully and have made correction which we hope meet PLOS ONE’s publication criteria. Revised portions are marked red in the paper. The main corrections in the paper and the responds to the journal requirements and editor’s comments are as following:

Journal Requirements:

Response: Thank you for bringing this to our attention. We have thoroughly reviewed the reference list to ensure its completeness and accuracy. We have also verified that no retracted papers have been cited in the manuscript. Your reminder is greatly appreciated, and we are committed to maintaining the highest standards of integrity in our research.

Additional Editor Comments:

Dear authors,

The revision are mostly satisfactory.

Minor comments

- Multiple grammatical errors still remains

- Dear authors,

Abstract

- The databases used for the review should be stipulated

Response: Thank you for your positive feedback and support for our study. We have taken note of your suggestions and have worked to improve the language, grammar, and overall clarity of the manuscript. We have also included the databases used in the review in the Abstract as recommended. We strive to ensure that the manuscript is error-free and meets the highest standards of quality. Thank you for your valuable input.

---

## [Editor Report · Decision Letter 3]

3 May 2024

Efficacy and safety of acupuncture for postpartum hypogalactia: A systematic review and meta-analysis of randomized controlled trials

PONE-D-23-25490R3

Dear Dr Fan

We’re pleased to inform you that your manuscript has been judged scientifically suitable for publication and will be formally accepted for publication once it meets all outstanding technical requirements.

Kind regards,

Benjamin Jun Jie Seng, MD

Academic Editor

PLOS ONE

Additional Editor Comments (optional):

Minor grammatical errors. Will leave it to the editorial office staff to proofread and discuss with the primary authors

---

## [Editor Report · Acceptance letter]

28 May 2024

PONE-D-23-25490R3 

PLOS ONE

Dear Dr. Liang, 

I'm pleased to inform you that your manuscript has been deemed suitable for publication in PLOS ONE. Congratulations! Your manuscript is now being handed over to our production team.

Kind regards, 

on behalf of

Dr. Benjamin Jun Jie Seng 

Academic Editor

PLOS ONE